# Periodic structural changes in Pd nanoparticles during oscillatory CO oxidation reaction

Tanmay Ghosh [1,2], Juan Manuel Arce-Ramos [3], Wen-Qing Li[3], Hongwei Yan[2], See Wee Chee [1,2], Alexander Genest [3,4] & Utkur Mirsaidov [1,2,5,6] ✉

Nanoparticle (NP) catalysts are ubiquitous in energy systems, chemical production, and reducing the environmental impact of many industrial processes. Under reactive environments, the availability of catalytically active sites on the NP surface is determined by its dynamic structure. However, atomic-scale insights into how a NP surface reconstructs under reaction conditions and the impact of the reconstruction on catalytic activity are still lacking. Using operando transmission electron microscopy, we show that Pd NPs exhibit periodic round–to–flat transitions altering their facets during CO oxidation reaction at atmospheric pressure and elevated temperatures. This restructuring causes spontaneous oscillations in the conversion of CO to $CO_2$ under constant reaction conditions. Our study reveals that the oscillatory behavior stems from the CO-adsorption-mediated periodic restructuring of the nanocatalysts between high-index-faceted round and low-index-faceted flat shapes. These atomic-scale insights into the dynamic surface properties of NPs under reactive conditions play an important role in the design of high-performance catalysts.

The surface characteristics of a heterogeneous catalyst control its interaction with adsorbates and drive the reaction cycle[1–4] in many energy-related and environmental catalytic conversions[5]. Hence, understanding how the surfaces impact the activity is essential for a rational design of high-performance catalysts. In most practical applications, catalysts are used in the form of nanometer-sized particles[6,7]. Nanoparticle (NP) catalysts expose various facets to the reactants simultaneously, with each of them having surface atoms with different coordinative environments[6,8,9]. However, when the NPs are placed under reaction conditions, they will evolve according to the environmental parameters (i.e., gas composition, pressure, and temperature) and adopt new catalytically active structures[2,3,10–12]. For example, molecules adsorbed from the gas phase can change the free energy of exposed surfaces of NPs and lead to an atomic rearrangement and the formation of new surface sites[13–16]. These changes in surface structure, in turn, modify their catalytic activity[2].

The adsorption and oxidation of CO molecules over noble metals, e.g., Pt, Pd, and Rh, are the basic surface processes that are often used to probe heterogeneous catalysis[17–20]. So far, the vast majority of operando studies focus on using bulk single crystal model catalysts to extrapolate the key structure–property relationships[1,21–27]. However, it is unclear how these insights translate into the relevant catalytic behavior of NP catalysts that are used in current technologies[3,7,11,28]. Hence, there are significant efforts in trying to understand the dynamical structure of a NP's surface during reactions with atomic-scale details[3,4,29]. In that respect, operando transmission electron

[1]Department of Physics, National University of Singapore, Singapore 117551, Singapore. [2]Centre for BioImaging Sciences, Department of Biological Sciences, National University of Singapore, Singapore 117557, Singapore. [3]Institute of High Performance Computing, Agency for Science, Technology and Research, Singapore 138632, Singapore. [4]Institute of Materials Chemistry, Technische Universität Wien, Getreidemarkt 9/BC, 1060 Vienna, Austria. [5]Centre for Advanced 2D Materials and Graphene Research Centre, National University of Singapore, Singapore 117546, Singapore. [6]Department of Materials Science and Engineering, National University of Singapore, Singapore 117575, Singapore. ✉e-mail: mirsaidov@nus.edu.sg

microscopy (TEM) is one of the most powerful techniques for visualizing the structure of individual NPs at high spatial resolution and under reaction conditions[2–4,29]. For example, it is possible to visualize the formation of high-index steps on the {100} surface facets of a Pt NP in the presence of CO[14]. Integration of mass spectrometry and nanocalorimetry enables us to probe the catalytic performance while imaging the dynamic morphology of a working NP catalyst[2,3].

One of the most critical phenomena in catalysis that determines the efficacy of the conversion reaction is oscillatory reactivity, where the catalyst switches back and forth between active and inactive states under constant reaction conditions[30–34]. These oscillations have been attributed to the combined effect of non-equilibrium states (i.e., heat and mass transfer between a catalyst and its surroundings) and repetition of reaction steps in a catalytic cycle[30,31,33]. It is known that noble metal (e.g., Pd, Pt, and Rh) catalysts can show periodic oscillations during CO-to-$CO_2$ conversion, alternating between high and low levels of activity[3,30,31]. Studies using single-crystal surfaces have ascribed the periodic transformation in conversion rate to the dynamic changes in the catalyst surface[30,31,35–38]. Moreover, a recent operando TEM study of oscillatory reaction in Pt NPs by Vendelbo et al. directly shows that structural changes and reactivity are correlated[3]. Yet, the mechanisms by which such oscillations occur are hotly debated[3,39–43].

Here, using cubic and octahedral Pd NPs, which have predominantly {100} and {111} facets, respectively, we show that high-index facets periodically emerge and disappear on these NP surfaces concurrently with the periodic transition from high- to low-activity states that causes the oscillation in reactivity. Contrary to the existing opinions[31,36,37,44] that the oscillations should only appear under $O_2$-rich conditions, we observe oscillations also under CO-rich conditions. We show that periodic restructuring of Pd NP catalysts is the root cause of oscillation in reactivity for both $O_2$- and CO-rich cases, and these oscillations are unlikely to be caused by the oxidation of the surface, as previously proposed[37]. These observations are supported by density functional theory (DFT) and thermodynamic calculations. There is a robust correlation between calculated structures and experimentally imaged NPs. These results provide direct evidence of how CO-coverage-mediated structural rearrangement can modulate the active sites on the surface of a Pd NP at constant temperatures, which cannot be observed outside of reaction conditions.

## Results and discussion

### Direct observation of the morphology and reactivity of NPs

Here we used an in situ gas-phase TEM imaging platform where the shape-controlled Pd NPs were encapsulated within a microfabricated gas cell with a thin film heater (Supplementary Fig. 1a). Previously, we showed that this approach can be used to follow changes in the structure of Pd, Pt and Rh NPs during heating and cooling in a reactive gas environment[2]. Our experiments with randomly shaped Pd NPs indicated that it is possible to observe oscillatory reactivity. Supplementary Fig. 1b–d show an example of an experiment where repeated fluctuations were seen in both the heater power (Supplementary Fig. 1c) and the corresponding changes in CO, $O_2$, and $CO_2$ partial pressures under a constant reaction temperature (Supplementary Fig. 1d). It was, however, difficult to identify the morphological origin of these oscillations due to the irregular shape of these NPs. To understand how oscillation correlates with morphology, we used shape-controlled NPs (Supplementary Fig. 2) and monitored structural changes, reaction temperature, and heater power, and also output gas compositions at constant reaction conditions. The TEM image series were captured using the electron counting mode of a direct electron detection camera at an optimized electron flux of <100 e⁻ Å⁻² s⁻¹ to minimize the effects that the electron illumination may have on the reaction while maintaining the lattice resolution. We dropcasted the chemically synthesized either cubic or octahedron Pd NPs (after removing surfactants[45]) over the heater area. To remove any residual

surfactant, we first heated the NPs at 300 °C under Ar followed by heating under an oxygen gas environment, 20% $O_2$ and 80% He, at 300 °C[46–48]. Next, we introduced a gas mixture of $O_2$ (diluted by He) and CO into the microfabricated gas cell and studied the catalytic and surface structural properties of the Pd NP catalysts at various reaction conditions (i.e., different temperatures and $p_{CO}/p_{O_2}$ pressure ratios).

First, we tested the impact of $O_2$-rich conditions (i.e., a mixture of 10% CO, 18% $O_2$, and 72% He, which corresponds to $p_{CO}/p_{O_2} \approx 0.5$) on the morphology and reactivity of Pd nanocubes and nano-octahedrons at various constant temperatures by tracking these NPs and simultaneously monitoring the CO, $O_2$, and $CO_2$ output signals. Scanning TEM (STEM) and TEM images of a nanocube and nano-octahedron and changes in gas compositions during their reactions at different temperatures are shown in Fig. 1. These observations reveal that nanocubes and nano-octahedrons respond differently to the changes in the environment. Nanocubes (with {100} facets) were inactive up until 400 °C, and only then showed a small increase in CO conversion as we gradually increased the temperature to 460 °C (Fig. 1c). On the other hand, nano-octahedrons (with {111} facets) became active at 360 °C and immediately reached full activity (Fig. 1d). This low-temperature activity of nano-octahedron is due to the lower activation energy of {111} surface for CO oxidation as compared to the {100} surface[49,50].

TEM images of the nanocubes taken at different reaction temperatures also showed no significant structural changes (Fig. 1a). In contrast, the corners of the nano-octahedrons were found to be flat at 300 °C and round at 360 °C (Fig. 1b), which is well above their ignition temperature of 340 °C, i.e., the temperature above which a catalyst becomes active (Supplementary Fig. 3). More strikingly, we found that the activity of the nano-octahedrons periodically oscillates between high- and low-activity steady states at fixed reaction temperatures above the ignition (e.g., at 360 °C). Concurrently, there were periodic up and down spikes in measured temperature (Fig. 1d) coinciding with the oscillations in $CO_2$ production, consistent with the respective high- and low-activity states of the catalysts (see Supplementary Fig. 4 for the enlarged version of Fig. 1d). Nanocubes did not exhibit such oscillations (Fig. 1c). This trend was confirmed throughout multiple experiments (Supplementary Fig. 5).

Next, we examined the oscillation of nano-octahedrons under $O_2$-rich conditions in detail. Figure 2 shows the plot of the measured temperature and compensated heater power (Fig. 2a) along with mass spectrometry signals displaying the oscillatory pattern in $CO_2$ production (Fig. 2b) at 380 °C and a fixed partial gas pressure ratio of $p_{CO}/p_{O_2} \approx 0.5$. In situ TEM image series tracking the structure of a nano-octahedron during the oscillation reveals that the corners of the nano-octahedron are flat at the low-activity state (e.g., at 0 and 545 s) and round at the high-activity state (e.g., at 330 and 765 s), and these corners periodically restructure (i.e., −flat−round−flat−) concurrent with the reaction rate oscillations (−low−high−low−) (Fig. 2c and Supplementary Movie 1). The high-resolution TEM images of the corners in Fig. 2d show that the surface of a low-activity nano-octahedron is terminated with low-index {100}, {110}, and {111} facets (at 0 and 545 s), whereas the restructured surface of a high-activity nano-octahedron is round, indicating the presence of high-index facets, e.g., {120} between the {100} and {110} facets or {311} between the {100} and {111} facets (at 330 and 765 s). These results clearly suggest that the reaction rate transition is correlated to the surface restructuring of a NP catalyst during the oscillation. Note that both cubic and octahedral NPs retain their original shape even after the conversion reactions that last couple of hours, indicating that all structural changes occurring during the reaction are transient (see Supplementary Figs. 6, 7, and 16).

Since nanocubes with predominantly {100} facets do not exhibit oscillations, it appears that the presence of a {111} facet is crucial for such behavior. To test this hypothesis, we formed truncated Pd nanocubes with {111} facets at their corners by heating the nanocubes at 300 °C under $O_2$ environment (20% $O_2$ and 80% He) (Fig. 3c and

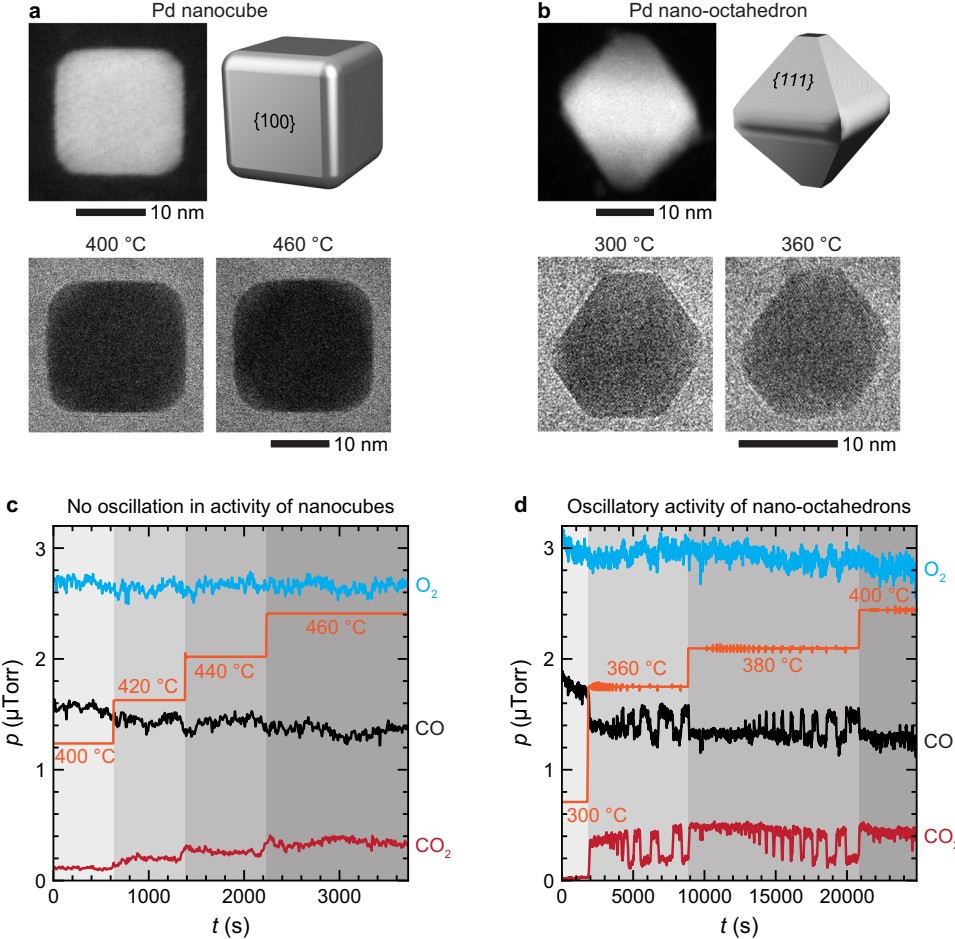

**Fig. 1 | Operando transmission electron microscopy (TEM) of Pd nanocube and nano-octahedron catalysts during a CO oxidation reaction.** STEM and TEM images of **a** cuboidal and **b** octahedron Pd nanoparticles (NPs) at different temperatures and under 760 Torr of 10% CO, 18% $O_2$, and 72% He gas environment corresponding to a gas pressure ratio of $p_{CO}/p_{O_2} \approx 0.5$. Changes in gas compositions during the CO oxidation reaction over **c** nanocubes and **d** nano-octahedrons at different temperatures as indicated by orange step curves. The reaction oscillates for the nano-octahedrons, while the nanocubes do not show such oscillatory reactivity. In terms of the NP morphology, the sharp corners of nano-octahedrons in **b** become rounded as they switch to a high-activity state at 360 °C, whereas the nanocubes in **a** remain unchanged, and their activity increases very little at elevated temperatures.

Supplementary Fig. 8). As expected, under $O_2$-rich conditions, these truncated nanocubes also exhibited spontaneous oscillations above their ignition temperatures (Fig. 3) (for more examples, see Supplementary Figs. 9–15). The oscillatory wave patterns of the measured temperature and heater power coincide with the oxidation rates of CO to $CO_2$ (Fig. 3b) in a manner consistent with an exothermic reaction (Fig. 3a, b). More importantly, similar to the nano-octahedrons (Fig. 2c, d), these oscillations in the reactivity are also consistent with the periodic restructuring of the NP, as seen from the TEM image series in Fig. 3d, e and Supplementary Figs. 10 and 11 (Supplementary Movie 2). High-resolution image series in Fig. 3e suggest that the low-activity truncated nanocube is {111}, {110}, and {100} facet-dominant (e.g., at 102 and 326 s), whereas steps and edges are prevalent in high-activity states (e.g., at 204 s), indicating the presence of high-index facets (e.g., {120} and {311}). These observations further confirm that the presence of {111} is crucial for exhibiting an oscillatory behavior, and the reaction rate transitions are correlated to the surface restructuring of a NP catalyst.

To test how the reactivity of truncated nanocubes changes with the reaction temperature, we carried out CO oxidation reactions at four different temperatures (400, 420, 440, and 460 °C) at $p_{CO}/p_{O_2} \approx 0.5$ (Supplementary Fig. 17). In all the cases, the oscillations were present (Supplementary Fig. 17a–d), and the frequency of these oscillations increases with an increase in temperature while

the amount of heat released per cycle decreases (Supplementary Fig. 17e, f), indicating a shift towards a fully active state at higher temperatures.

To verify that all the NPs take part in surface restructuring during an oscillation, we also captured a lower magnification movie with six to eight NPs in the field of view. The in situ TEM image series show that all the NPs restructure at the same time in a synchronized manner, accompanied by the periodic transition of low- to high-activity steady states (Supplementary Figs. 18 and 19), clearly indicating that structural oscillations are synchronized across different NPs. The synchronized structural oscillations indicate that the NPs are coupled through gas-phase mass transfer, as suggested by earlier reports for a continuous stirring tank reactor[41].

## Impact of CO amount on surface restructuring and activity switching of NPs during oscillation

While the above results establish that there is a clear correlation between surface restructuring and activity switching during the oscillations, we need to address the question regarding the dominant driving factors that cause the restructuring. Earlier bulk studies had attributed these reaction oscillations to the oscillatory transformation of a catalytic surface between a metallic and oxide one[37,38,44]. However, in our experiments, we see no evidence for such metal-to-oxide transformation during the oscillations under various reaction

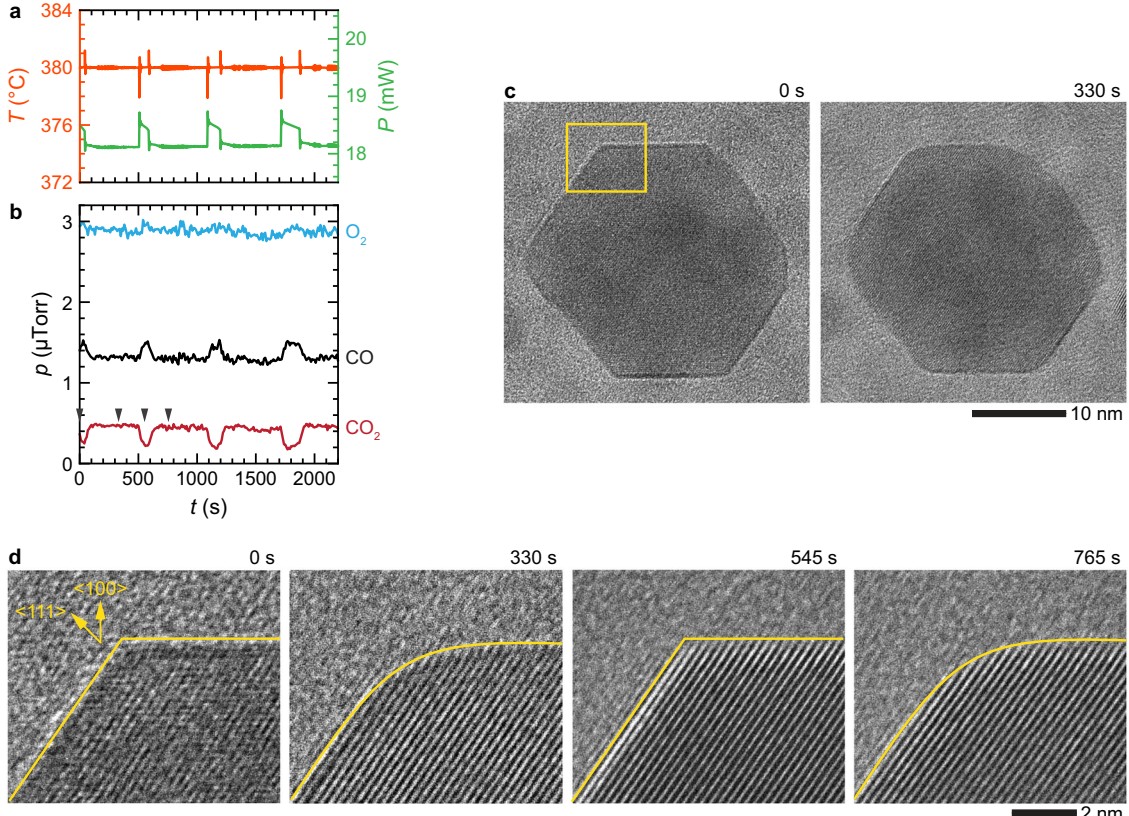

**Fig. 2 | Restructuring of a Pd nano-octahedron during an oscillatory CO oxidation reaction. a** Plots of the measured temperature (orange curve), heater power (green curve), and **b** the corresponding amounts of the CO (black curve), $O_2$ (blue curve), and $CO_2$ (red curve) gases during an oscillatory CO oxidation reaction at 380 °C and $p_{CO}/p_{O_2} \approx 0.5$. The arrows in **b** correspond to the timepoints of the

image series shown in **c**, **d**. **c** In situ TEM image series of a nano-octahedron in low- and high-activity states during the oscillation. **d** High-resolution image series of the NP's corner, as indicated by the yellow box in **c**. During the reaction, the corners of the nano-octahedron periodically restructure between high-activity round (330 and 765 s) and low-activity flat (0 and 545 s) facets (Supplementary Movie 1).

conditions that range from $O_2$-rich to CO-rich atmospheres (Fig. 4 and Supplementary Figs. 20 and 22).

Figure 4a–c displays how the reactivity and surface restructuring of a truncated nanocube change when the gas ratio, $p_{CO}/p_{O_2}$, increases from 0.8 to 2.0, at a constant temperature of 460 °C ($t = 0$–1400 s). The TEM image series recorded at a gas ratio of $p_{CO}/p_{O_2} \approx 0.8$ reveals that the periodic restructuring of the corners of the nanocube between low-activity flat and high-activity round ones occurs concurrently with reaction oscillation (Fig. 4a–c and Supplementary Movie 3), similar to the case of the gas ratio of $p_{CO}/p_{O_2} \approx 0.5$ shown in Fig. 3. While we observed regular oscillations at a gas ratio of $p_{CO}/p_{O_2} \approx 0.8$ ($t = 0$–1000 s, in Fig. 4a, b), the oscillation stopped as soon as the CO pressure was doubled (with a gas ratio of $p_{CO}/p_{O_2} \approx 2.0$), and the catalyst became inactive, as suggested by the flattened curves of temperature, heater power, and $CO_2$ signal (Fig. 4a, b, $t = 1000$–1400 s). Also, when the gas ratio was increased to $p_{CO}/p_{O_2} \approx 2.0$, the round corners of the nanocube transformed to flat concurrently with the quenching in the activity (Fig. 4a–c, $t = 1000$–1400 s).

Next, we ramped up the temperature to 540 °C while keeping the gas ratio fixed at $p_{CO}/p_{O_2} \approx 2.0$. We found that the ignition temperature had shifted upwards to 522 °C, where $CO_2$ production spiked again (Fig. 4a, b, $t = 1480$ s). This upward shift in the ignition temperature with increasing CO partial pressure is consistent with earlier reports[2,19,20]. At 540 °C, the catalyst again displayed the reaction oscillations at ~110 s after the ignition ($t = 1590$–2200 s). During the transition from 460 to 540 °C, we recorded a movie of the same nanocube shown in Fig. 4c. The TEM image at 1480 s shows that the flat corners of the nanocube became rounded when the temperature increased from 460 to 540 °C concurrently with a spike in $CO_2$ signal

(Fig. 4b, c). At 1590 s, the round corners became flat again concurrently with a decrease in $CO_2$ signal, even though the reaction conditions remained the same. Eventually, reaction and structural oscillations were established. These results show that the reaction rate oscillation can arise under CO-rich conditions and is associated with the periodic restructuring of NP catalysts. The arising of oscillations in CO-rich conditions and its correlation with surface restructuring emphasize that CO-mediated periodic restructuring of Pd NPs is the general driving force that causes reaction oscillations in both $O_2$- and CO-rich environments.

A careful examination of the NP surface in Fig. 4c reveals the presence of a low-contrast amorphous material, which we attribute to carbonaceous residues left from the NPs synthesis because of the following two reasons. First, these residues were only observed on rare occasions (for example, none of the NPs shown in Supplementary Figs. 18 and 19 display any surface residues). Second, the structure of residues did not change during the entire reaction. Nevertheless, to ensure that these surface materials are not PdO, which might have formed during the reaction[19,20], we first carefully oxidized the surface of the Pd nanocubes (in 20% $O_2$ and 80% He atmosphere at 460 °C) and found that the image contrast from the crystalline surface PdO (Supplementary Fig. 21b) is much stronger than from the surface residues in Fig. 4c. Moreover, the oxide reduces and vanishes under the typical reaction condition ($p_{CO}/p_{O_2} \approx 0.8$ at 460 °C) (Supplementary Fig. 21c). In any case, because PdO rapidly reduces into metallic Pd in the presence of CO (Supplementary Fig. 21c), any initial surface oxide that may be present on the surface of our Pd NPs does not affect their reactivity (i.e., the NPs retain their oscillatory reactivity) (Supplementary Fig. 22).

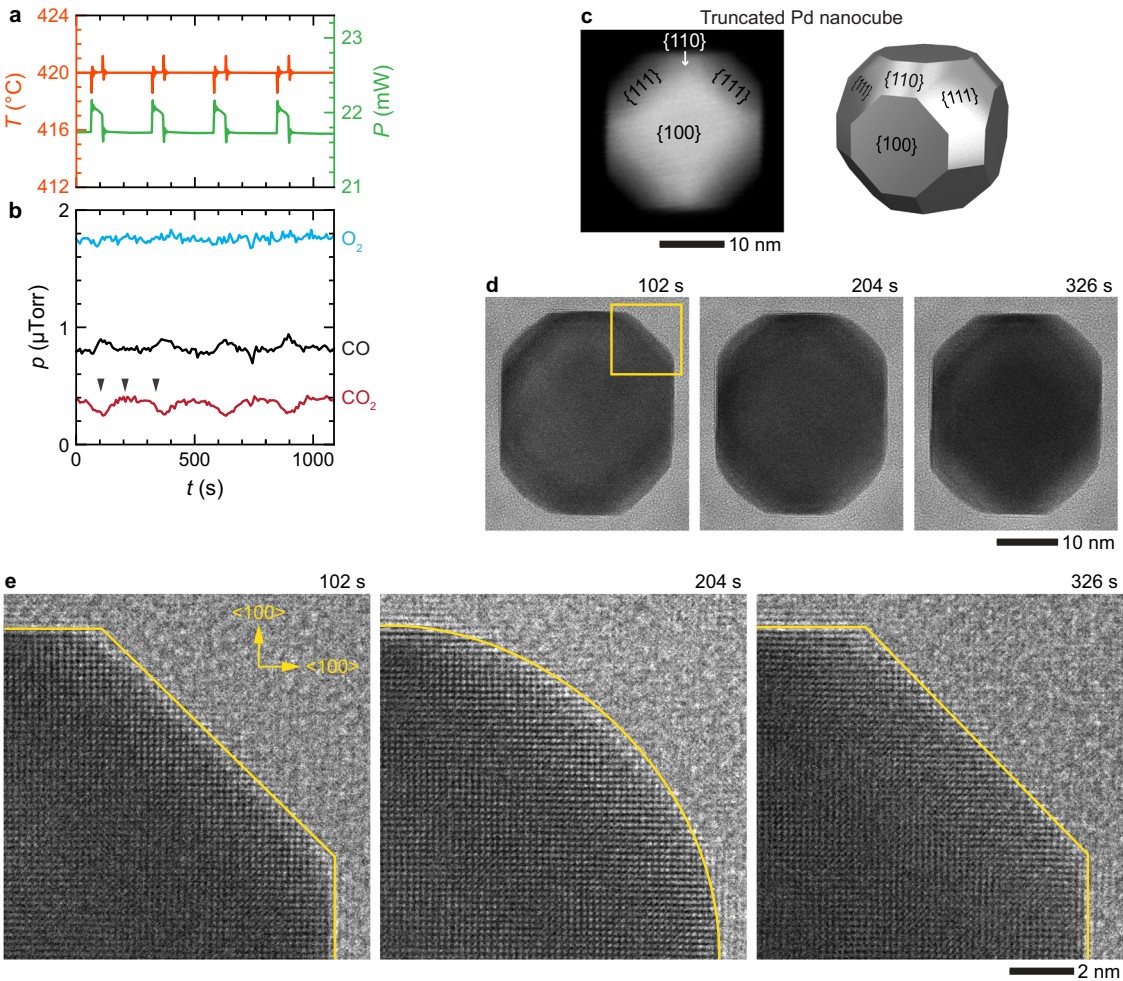

**Fig. 3 | Restructuring of a truncated Pd nanocube during an oscillatory CO oxidation reaction. a** Plots of the measured temperature, heater power, and **b** the corresponding amounts of the CO, $O_2$, and $CO_2$ gases during the oscillatory CO oxidation reaction at 420 °C and $p_{CO}/p_{O_2} \approx 0.5$. The arrows in **b** correspond to the timepoints of the image series shown in **d**, **e**. **c** STEM image and the schematic of a truncated nanocube. **d** In situ TEM image series of a truncated nanocube in low- and high-activity states during the oscillation. **e** High-resolution image series of the NP's corner, as indicated by the yellow box in **d**. During the reaction, the corners of the truncated nanocube restructure between the round (at 204 s) and flat (at 102 and 326 s) facets (Supplementary Movie 2; see Supplementary Fig. 10 for the details of the rest of oscillation during 500 s < $t$ < 700 s).

Note that with the shift to CO-rich conditions and the corresponding increase in ignition temperature (Fig. 4d), the amount of heat released during exothermic CO oxidation reaction increases, as seen by the increase in amplitudes of ignition-induced temperature jump ($\delta T_{ig}$) and the jump in temperature ($\delta T$) at the onset of high-activity states of the oscillatory reaction (Fig. 4e, f). Furthermore, the detailed analysis of reaction oscillations reveals that the oscillation frequency increases with an increase in $p_{CO}/p_{O_2}$ ratio, while the dwell time of the high-activity state decreases (Fig. 4d and Supplementary Fig. 20).

### DFT and thermodynamic calculations

To understand the effect of constant reaction conditions on the observed periodic restructuring of the Pd NPs, we modeled 13–15 nm Pd NPs using ab initio thermodynamic calculations and a Wulff construction[51] at experimentally used reaction conditions (Section 1 in Supplementary Note). The transient nature of the NPs during the reactions indicates a swift response to external driving forces, which we aim to rationalize by focusing on the endpoints of this process, i.e., equilibrium states provide accurate NP models. Here, we considered high-index facets {120} and {311} situated at the {100}–{110} and {100}–{111} boundaries, respectively, in addition to the {100}, {110}, and {111} low-index facets. By

evaluating the CO and O coverage-dependant surface energy of the various facets at specific conditions, we estimated the shape of the resulting NP (Sections 2–3 in Supplementary Note and Supplementary Figs. 23–29). We assessed the change in NP shape by inspecting the varying coordination numbers (CNs) of surface atoms, which may be used to trace the fraction of surface atoms belonging to specific facets (Fig. 5a and Supplementary Fig. 30). The most prominent change in the shape of the NPs with temperature corresponds to the decrease in proportions of the {110} facet fraction with the concomitant increase in {111} presence, as indicated by changes in the fraction of surface atoms with CNs of 7 and 11 (from {110} facets), and CN of 9 (from {111} facets) (Fig. 5a). With a partial pressure ratio of $p_{CO}/p_{O_2} = 1.0$, the fraction of surface atoms with CNs of 7 and 11 drops from 0.49 and 0.46 to 0.07 and 0.01, respectively, in the temperature range of 290–335 °C, while the fraction of atoms with CN of 9 simultaneously increases from 0.00 to 0.80. Our calculations show that the {110}-facet-dominated NP restructures to the one where the dominant facets are {111} at higher temperatures for the reactive atmospheres with elevated $p_{CO}/p_{O_2}$ pressure ratios. The {110} facets disappear from the calculated NPs at 305, 335, 350, and 365 °C for pressure ratios of 0.5, 1.0, 1.5, and 2.0, respectively (Supplementary Fig. 31). This is consistent with the

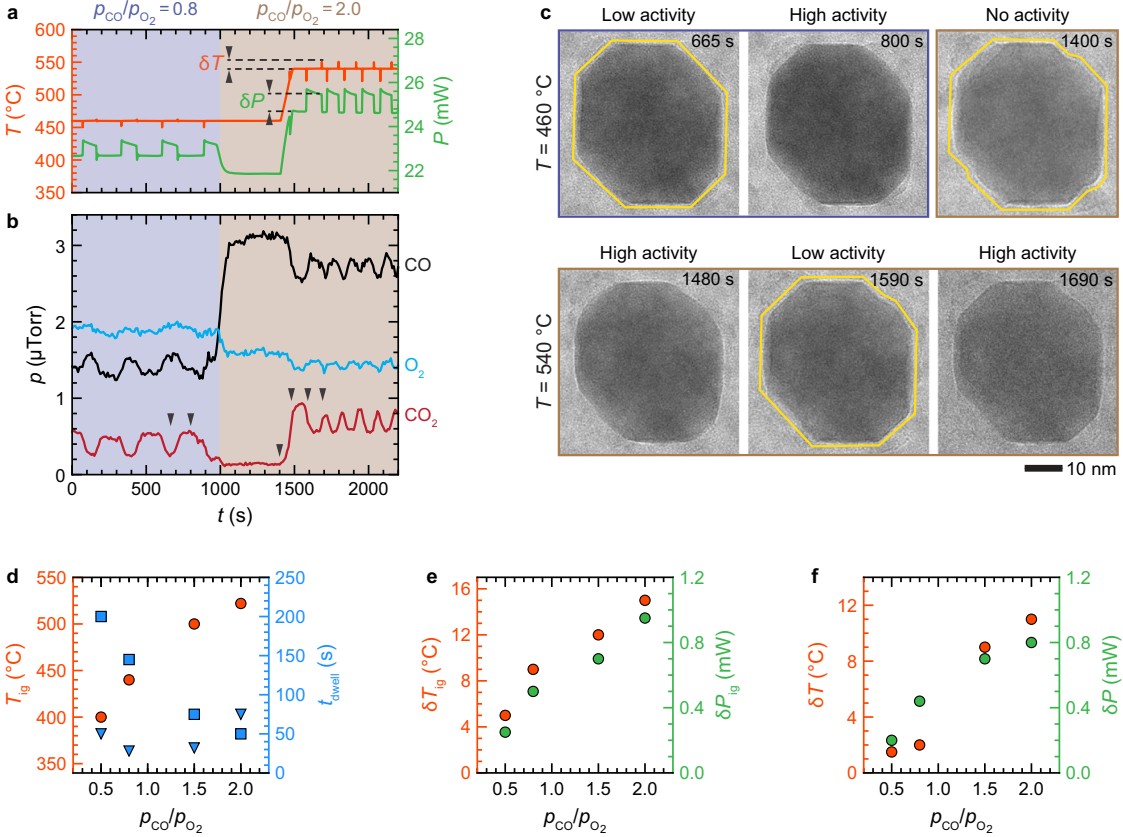

**Fig. 4 | Impact of CO pressure ($p_{co}$) on the oscillatory reaction over truncated Pd nanocube. a** Plots of the measured temperature, heater power, and **b** the corresponding amounts of the CO, $O_2$, and $CO_2$ gases during the CO oxidation reaction at 460 °C and $p_{CO}/p_{O_2} \approx 0.8$ and at 540 °C and $p_{CO}/p_{O_2} \approx 2.0$. The arrows in **b** correspond to the timepoints of the image series shown in **c. c** In situ TEM image series of a truncated nanocube showing a periodic restructuring corresponding to the oscillations during $t = 0$–1000 s at $p_{CO}/p_{O_2} \approx 0.8$ (Supplementary Movie 3). At $t = 1000$–1400 s, when the partial pressure was increased to $p_{CO}/p_{O_2} \approx 2.0$, while maintaining the temperature at 460 °C, the corner of the nanocube became flat, and the activity dropped almost to 0. The oscillation resumed under this CO-rich condition ($p_{CO}/p_{O_2} \approx 2.0$) when the temperature was

increased to 540 °C at $t \geq 1590$ s (Supplementary Movie 4). **d** The ignition temperatures for the conversion reaction ($T_{ig}$, orange circles) as a function of $p_{CO}/p_{O_2}$ and the corresponding dwell time of catalyst in low-activity (blue triangles) and high-activity (blue squares) states. **e** Ignition-induced temperature jump ($\delta T_{ig}$, orange circles) and concurrent drop in heater power ($\delta P_{ig}$, green circles), as indicated by arrows in Supplementary Fig. 20d as a function of the gas ratio ($p_{CO}/p_{O_2}$). **f** The jump in temperature ($\delta T$, orange circles) and concurrent drop in heater power ($\delta P$, green circles) as a function of the gas ratio ($p_{CO}/p_{O_2}$) at the onset of high-activity states of the oscillatory reaction, as described in **a**. The results shown in **d**–**f** are extracted from the measurements shown in Supplementary Fig. 20.

experimentally established upward shift in the ignition temperature under CO-rich conditions (Fig. 4d).

As the {110} facets start to disappear from the calculated NPs, the high-index facets become visible. Figure 5b shows that {120} and {311} high-index facets emerge in our modeled NP surfaces when the temperature increases by 10 °C, from 320 to 330 °C, at $p_{CO}/p_{O_2} = 1.0$. This indicates that such a difference in temperature is sufficient to produce the restructuring of the Pd NP. In our experiments, we observed a jump in temperature ($\delta T_{ig}$) (which ranges from 5 to 15 °C, depending on $p_{CO}/p_{O_2}$) at the time of ignition, at each of $p_{CO}/p_{O_2}$ pressure ratios (Fig. 4e). We anticipate that this ignition-induced jump in temperature could initiate such a structural oscillation, where high-index facets can periodically emerge and disappear at the corners of low-index facets, as shown in Figs. 2d and 3e.

The link between the structural changes and oscillations in activity under constant reaction conditions can be understood by recognizing that the metal surface rarely consists of both CO and O adsorbates, which are required for the reaction to proceed[19]. The Pd surface is, in general, dominated by adsorbed CO molecules regardless of the gas composition, as CO binds more strongly than O[52,53]. Once light-off occurs, the energy released by CO oxidation will sustain the reaction, leading to ignition[2]. However, this is true only if CO molecules can combine with O to produce $CO_2$. Because the binding strength of CO is

different on different facets, we can reasonably assume that any reaction will initiate at the corners of a catalyst particle since they are expected to be the high-activity sites[52,53]. However, the reaction cannot proceed further if the lower index facets remain poisoned by CO. In this case, the subsequent conversion will be quenched, and the exposed catalyst surface gets repopulated with CO molecules, thereby setting up the oscillatory phenomena. This explanation is also consistent with the increase in oscillation frequency with increasing temperature (Supplementary Fig. 17), where the desorption of CO becomes favorable, creating sites for the reaction with O (Supplementary Figs. 32 and 33).

Furthermore, we calculated the temperature spans ($\delta T_{span}$) at which the NP shape starts to change from {110}-facet-dominated to a one without {110} facets, at the $p_{CO}/p_{O_2}$ ratios of 0.5, 1.0, 1.5, and 2.0 (Section 3, in Supplementary Note) and compared them with the experimental measurements. The calculated $\delta T_{span}$ widens with increasing $p_{CO}/p_{O_2}$ ratio (Fig. 5c and Supplementary Fig. 31). This trend can be understood by analyzing the temperature-dependent CO coverage over the various facets, $\theta_{CO}$, and its changes at different $p_{CO}/p_{O_2}$ ratios (Supplementary Fig. 32), but the rate at which $\theta_{CO}$ drops with temperature decreases (Supplementary Fig. 33). These changes in $\theta_{CO}$, which affect the surface energy of the NP (Eq. 1), are

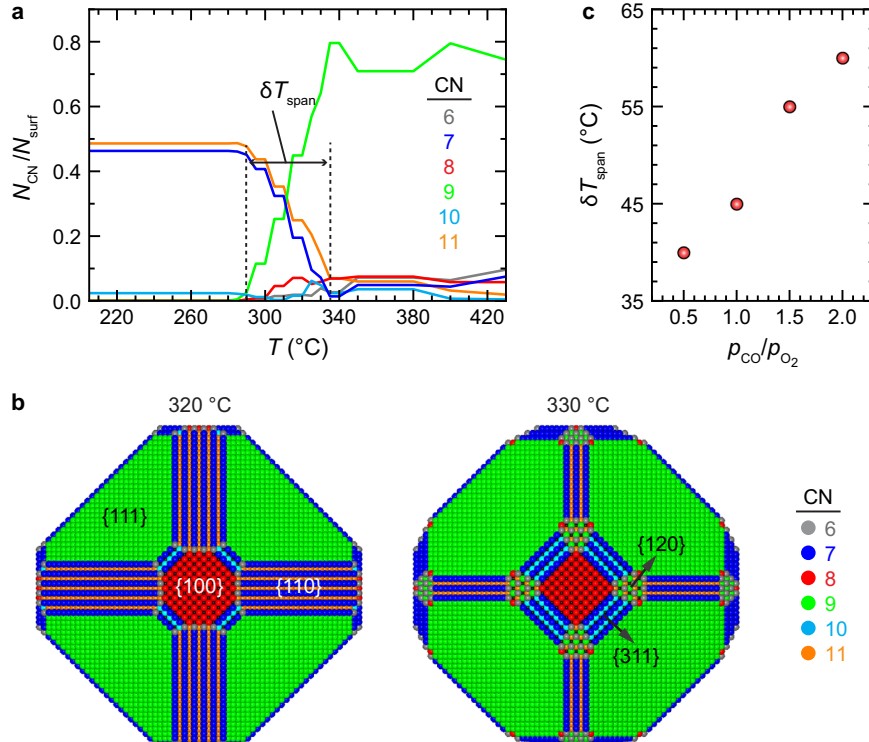

**Fig. 5 | Modeling Pd NPs under reaction conditions. a** Fraction of surface atoms with various coordination numbers (CNs) in Pd NPs, $N_{CN}/N_{surf}$, generated at $p_{CO}/p_{O_2} = 1.0$, with $O_2$ partial pressure ($p_{O_2}$) of 140 Torr. $N_{surf}$ is the total number of surface atoms. Each colored line corresponds to the fraction of surface atoms with a different CN. **b** Snapshots of the NP at 320 and 330 °C. The atoms were colored according to their coordination number, as indicated in the legend next to the NPs. The diameter of modeled NPs is between 13 and 15 nm and they comprise around 60,000 atoms. **c** The thermodynamically calculated temperature spans ($\delta T_{span}$) where {110} dominant NP shape is transformed to {111} dominant one as a function of $p_{CO}/p_{O_2}$ (Fig. 5a and Supplementary Fig. 31).

why the shape of the NP changes under different reaction conditions. The positive correlation between $\delta T_{span}$ with $p_{CO}/p_{O_2}$ ratio means that at higher $p_{CO}/p_{O_2}$ ratios, the amount of heat released during a periodic restructuring of the NP occurring concurrently with the high- to low-activity switching would be greater. This is consistent with our experimental observations, where the amplitude of temperature jump ($\delta T$, in Fig. 4a) increased from 1.5 to 11 °C when $p_{CO}/p_{O_2}$ ratio increased from 0.5 to 2.0 (Fig. 4f).

Although the oscillations in CO oxidation over bulk Pd catalysts are attributed either to the formation of subsurface oxide layer[32,36] (at low reaction pressures) or to the surface oxides[37,44] (at high reaction pressures), the role of the surface restructuring was largely overlooked. Our direct observations, together with the DFT calculations, reveal that the oscillations are the direct consequence of coupled periodic restructuring of the NPs' surface between the high-index-faceted active state and low-index-faceted inactive state. While the oxide layers may still contribute to the oscillatory activity in a subtle way, our observations clearly show that oscillations in the activity and structure of the NPs occur in tandem. Furthermore, our calculations show that observed oscillatory reactivity both under $O_2$-rich and $O_2$-poor conditions is consistent with the periodic restructuring of NPs. Hence, the oscillatory reactivity is a direct consequence of the periodic restructuring of the nanocatalysts' surface.

In conclusion, our combined operando TEM experiments and theoretical calculations unambiguously establish that there is a morphological origin of the reaction oscillation over Pd nanocatalysts. Observed periodic restructuring of Pd NPs between low-activity flat and high-activity round structures during CO oxidation stems from the difference in CO coverage and subsequent modification to the surface energy of different facets. We anticipate that this dynamic structure-dependent activation–deactivation behavior in Pd NPs at constant reaction conditions extends to many other catalyst systems. More generally, our approach to studying the interaction between catalyst surface and reactants with the sub-nanometer resolution is critical for identifying and eliminating different deactivation modes and will be important in guiding the design of future high-performance catalysts.

## Methods
### Synthesis of Pd nanocube and nano-octahedrons

The following reagents were used to synthesize the Pd NPs used in this study: diethylene glycol (purity of 99%, Cat. No. H26456, Sigma-Aldrich Co., St Louis, MO, USA), sodium tetrachloropalladate (II) (purity of 98%, Cat. No. 205818, Sigma-Aldrich Co., St Louis, MO, USA), hexadecyltrimethylammonium bromide (CTAB, purity of 99%, Cat. No. H9151, Sigma-Aldrich Co., St Louis, MO, USA), palladium(II) chloride (purity of 99.999%, Cat. No. 323373, Sigma-Aldrich Co., St Louis, MO, USA), polyvinylpyrrolidone (PVP) with an average molecular weight of 10 kDa, (Cat. No. PVP10, Sigma-Aldrich Co., St Louis, MO, USA), and pure water (Cat. No. 320072, Sigma-Aldrich Co., St Louis, MO, USA).

We modified the method described by Niu et al. to synthesize the Pd nanocubes[54]. First, we dissolved 23 mg of CTAB in 5 mL water and added 250 μL of 10 mM $H_2PdCl_4$ solution, followed by heating the mixture at 100 °C. Next, 250 μL of 100 mM ascorbic acid was added to the heated solution to produce the nanocubes. After 2 min, 250 μL of the resultant solution containing nanocubes was mixed with 1.75 mL of water and centrifuged three times at 13,500 × g to remove the excess CTAB, and the nanocubes were resuspended in 250 μL of water. Then, the nanocubes solution was dropcasted onto a microfabricated chip of a gas flow cell with the heating element (DENSsolutions, Delft, Netherlands). To remove any residual CTAB, we first heated the dropcasted nanocubes to 300 °C for 30 min under Ar atmosphere, followed by heating at 300 °C for 10 min under 80% He and 20% $O_2$ gas

environment inside the gas flow cell. To convert these nanocubes into truncated nanocubes, we heated them for another 1 h under 80% He and 20% $O_2$ environment. The morphology of the nanocubes and truncated nanocubes were characterized by TEM and STEM imaging, and these results are shown in Supplementary Figs. 2a and 8. Note that after a prolonged CO oxidation reaction, we observe the truncation of nanocubes because of the $O_2$ in the gas mixture. For example, under the $O_2$-rich reaction condition ($p_{CO}/p_{O_2} \approx 0.5$), nanocubes transformed into truncated nanocubes after approx. 6 h (Supplementary Fig. 16) and started to display oscillatory reactivity which is not observed in nanocubes (Supplementary Fig. 17).

To synthesize Pd nano-octahedrons, 80 mg of PVP was mixed with 2 mL of diethylene glycol, and the solution was heated to 135 °C in an oil bath for 5 min. Meanwhile, 27.2 mg of sodium tetrachloropalladate (II) was added to 1.75 mL of diethylene glycol. Then, 1 mL of this solution was injected into the previous PVP solution and left to react for 4 h, at which point the color of the solution changed from dark yellow to black, indicating the formation of Pd nano-octahedrons. We then washed as-synthesized Pd octahedrons in acetone and resuspended them in water. Prior to TEM experiments, we dropcasted them onto a microfabricated chip of a gas flow cell. We characterized the as-synthesized Pd nano-octahedrons by TEM and STEM imaging, as described in Supplementary Fig. 2b.

## Operando TEM experiments

Operando TEM experiments were carried out following the methods described in our earlier paper[2]. The TEM image series were acquired in Thermo Fisher 300 kV Titan TEM equipped with a Gatan K2 IS direct electron detection TEM camera (Gatan Inc., Pleasanton, CA, USA). In these experiments, we optimized the imaging conditions such that the electron flux was kept below 100 $e^-$ $Å^{-2}$ $s^{-1}$ at all times to avoid electron beam-induced artefacts, which is lower than what is commonly used in similar high-resolution in situ TEM studies[22,23]. The TEM image series were acquired at a rate of 1 frame per second using a Digital Micrograph plugin within the Gatan Microscopy Suite (GMS).

Prior to each experiment, the NPs were first dropcasted onto a heater chip of a gas cell, and the heater area of the entire chip was surveyed via low magnification TEM imaging to verify that the NPs are uniformly dispersed within the heater area and that their amount is similar across different experiments. Then, the NPs are heated to 300 °C under a flow of Ar for 30 min to remove any residual surfactants, followed by heat treatment in 20% $O_2$ and 80% He for 10 min before the introduction of CO. Note that the gas pressure inside the gas flow cell was kept at 760 Torr both during heat treatment and subsequent experiments. Next, during the conversion reaction study, the gas composition was adjusted by changing the gas flow in individual mass flow controllers installed within the gas delivery system (DENSsolutions, Delft, Netherlands). Here, two gas sources were used, one of which was pure CO gas, and the other was a pre-mixed gas comprising 20% $O_2$ and 80% He. To increase the reaction output, the gas was introduced at a slow flow rate of 0.08–0.10 mL $min^{-1}$ into the gas cell. The inline gas analyzer (DENSsolutions, Delft, Netherlands) was connected to the holder outlet line, and the gas compositions were measured with a quadrupole mass spectrometer (Stanford Research Systems, Sunnyvale, CA, USA). The amount of gas going into the analyzer chamber was controlled by a leak valve so that the chamber pressure was maintained in the range of $10^{-5}$ Torr.

During an oscillatory reaction, the image series were recorded for up to 30 min, during which NPs would slowly drift. The drifts were manually corrected on the fly to keep them within the field of view and in focus. Individual frames in the image series were further drift-corrected with post-processing alignment. The images presented in the manuscript had been extracted from multiple experiments, each using a new gas cell. In all cases, the same type of NPs behaved consistently across the separate experiments.

## Image and data processing

All image processing algorithms for the analysis of in situ movies were written in Python 2.7[55] using the NumPy[56], OpenCV[57], HyperSpy[58], and matplotlib[59] libraries. First, the raw Gatan dm4 image files of each image sequence were converted into 8-bit images. Then, the image sequence was drift corrected using cross-correlation template matching[60].

The drift corrected images were rotated by an angle such that the NP is parallel to the edges of an image. For each image, the rotation created four blank corners. These blank corners were replaced by the corners of the original image. To make the transition between the corners and the rotated image look natural, we removed the pixels along the intersection between the corners and the rotated image. Then, these pixels were filled up by texture-based image inpainting[61]. Supplementary Fig. 34 shows few examples of original, rotated, and edge-filled images displayed in Figs. 2c, 3d, and 4c. The high-resolution images presented in Figs. 2d and 3e were obtained by summing the inverse fast Fourier transform image with the original image using the GMS to accentuate the lattice fringes.

## Calculations of equilibrium shapes for NPs

We determined the shape of Pd NPs under different CO and $O_2$ atmospheres by combining ab initio thermodynamic calculations and the Wulff construction scheme[51]. DFT calculations using the BEEF-vdW functional[62] were used to estimate the tension energies of the studied planes ($\gamma_{hkl}$) and corrected due to the adsorption of either CO or O resulting in the interface tension energy ($\gamma_{hkl}^{int}$). The relevant input for the Wulff construction of NPs is

$$\gamma_{hkl}^{int} = \gamma_{hkl} + \frac{\theta_{CO}^{hkl} E_{ads}^{hkl}(CO) + \theta_{O}^{hkl} E_{ads}^{hkl}(O)}{A_{at.}^{hkl}} \qquad (1)$$

here, $E_{ads}^{hkl}(CO)$ and $E_{ads}^{hkl}(O)$ are the calculated coverage-dependent adsorption energies of CO and O over the $\{hkl\}$ surface. $\theta_{CO}^{hkl}$ and $\theta_{O}^{hkl}$ are the facet-specific surface coverages of CO and O. $A_{at.}^{hkl}$ is the area per surface atom of an $\{hkl\}$ facet. The coverages of adsorbed CO and O were estimated using the Fowler-Guggenheim model of adsorption, which considers adsorbate–adsorbate interactions. This scheme has already been employed elsewhere[10,63,64]. For more details on these ab initio thermodynamic calculations and the models used in this study, see the Supplementary Note.

## Data availability

The data that support the findings of this study are available from the corresponding author upon reasonable request.

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

## Acknowledgements

This work was supported by the Ministry of Education of Singapore (MOE-T2EP10220-0003). Juan M. Arce and Wen-Qing Li acknowledge the support from A*STAR AME IAF-PP grant (Grant No. A19E9a0103). The computational work benefited from a generous allotment of resources from the A*STAR Computational Resource Centre (ACRC) and the National Supercomputing Centre (NSCC) in Singapore.

## Author contributions

T.G., S.W.C., and U.M. conceived the research, T.G. prepared the samples, performed the experiments. T.G. and U.M. analyzed the data. Y.H. performed the image processing. J.M.A.-R., W.-Q.L., and A.G. performed the DFT calculations and theoretical modeling. All authors contributed to the writing of the manuscript.

## Competing interests

The authors declare no competing interests.
