## [Peer Review File · Nature Communications]

Title: Periodic structural changes in Pd nanoparticles during oscillatory CO oxidation reactionREVIEWER COMMENTS

Reviewer #1 (Remarks to the Author):

Ghosh et al. presented an interesting study to explore the underlying mechanism of oscillatory behavior of CO oxidation on Pd nanoparticles. The in-situ experimental and the DFT-based multiscale simulations showed this phenomenon is induced by the reconstruction of Pd nanoparticles owing to the change of the adsorbates' coverage. The paper is well-written and the demonstration is very clear. I think this paper meets the criteria of Nature Communications once the authors clarify the following issues.

(1) The experiments indicated the nanocubes did not exhibit the oscillation behavior as the nano-octahedrons. And the simulation gives the equilibrium structure at the specific temperature. Once the nanocubes and the nano-octahedrons were exposed in the same conditions for long time, such as 400 °C, CO : O₂ = 0.5, would they transform to the similar geometries and exhibit the similar oscillatory behaviors?

(2) Why did the nanocubes not show the oscillation? Is the activity of nanocubes too high to show the oscillation? Are the low activity surfaces necessary for the oscillation?

(3) The oscillated CO oxidation is a collective phenomenon. Clearly different nanoparticles may stay at the different stage of the structure reconstruction. In this work, the authors give the observation of single nanoparticle. I wonder if this oscillatory behavior is persistent during long-time reactions or only appears at the beginning stage. After long time operations, will the nanoparticles synchronize their structure reconstructions to show the stronger oscillation, or offset the individual effects to show the weaker oscillation?

(4) Can the authors provide the near-surface lattice distance in Figure 3 to show the oxidation contents of Pd nanoparticle during the CO oxidation?

(5) Did the total pressure keep at 1 bar for all reactions?

Reviewer #2 (Remarks to the Author):

The paper addresses a very important topic in catalysis, that is, the nature of the active site in reaction conditions. In particular, the authors report pieces of evidence for the periodic structural changes in Pd NP during CO oxidation. They link the CO oxidation oscillatory behavior with periodic morphological changes of the NP. This is proved experimentally mostly by correlating operando-TEM with the evolution of CO/CO₂/O₂ during time. On one side, the paper is well written and reports very interesting results. Results are novel as they provide an alternative explanation of the behavior already reported in the literature. On the other side, I have some points to be clarified that should be taken into consideration before publication:

1/ the difference between the behavior of nanocube NP and nano-octahedron NP is related to the presence of (111) facets. As far as I understand from the paper, this is first considered as the explanation because it is known (page 3 lines 107-109) that the reaction has lower activation energy on (111) than on (100). However, later (page 6 lines 149-151) the oscillatory behavior of the octahedron NP is

explained via the formation of high index facets. I found this quite confusing and not coherent. A clarification is required.

2/ An important effect is attributed to CO coverage. Are there available any IR experiments to prove the relative presence of these adsorbates at the different T and CO/O₂ ratios?

3/ CO could bind differently on the different terminations thus inhibiting the reaction on the different facets at different T.

4/ A point not clear to me is the origin of the oscillation. It can only start from the NP, since the feed is being kept constant. Comment on it should be important to better clarify the phenomenon.

5/ It is of extreme importance to provide evidence that the experimental rig is not affected by mass/heat transfer limitations. These aspects should be reported in the paper (with details in the SI).

6/ the theoretical analysis is of interest and important to support the interpretation of the results. However, there are recent papers that show that Wulff construction is not always a good model (e.g., the system needs time to reach such equilibrium - *Cat. Sci. Tech.* 8 (2018) 3493 and *ACS Catal.* 10 (2020) 6149). Are the periods of the oscillations long enough to allow the NPs to reach their equilibrium condition? Comment on this would be of added value.

Reviewer #3 (Remarks to the Author):

This work studied the periodic round-to-flat transitions of Pd NPs during CO oxidation reaction by using operando transmission electron microscopy. They used shape-controlled NPs and monitored structural changes, reaction temperature and heater power, and output gas compositions at constant reaction conditions. It is an interesting work to see periodic structural changes in Pd nanoparticles during oscillatory CO oxidation although it is similar to what was observed for Pt nanoparticles during CO oxidation [Ref. 3 in the manuscript]. Nevertheless, the authors have conducted a systematic study in order to obtain insights into Pd catalyzed CO oxidation. However, to explain what happened in this reaction, the authors seem to ignore a fact that Pd was partially oxidized in such a high temperature [refer to Fig. 4C and attached videos, the materials with weaker contrast around Pd NPs should be PdOx], which makes the conclusion questionable. In my experience, at such a high temperature with O₂, the Pd should be oxidized. If the Pd NPs were oxidized, the conclusion that “periodic restructuring of NP catalysts is the root cause of oscillation in reactivity for both O₂- and CO-rich cases and not caused by the oxidation of the surface as previously proposed” and the modeling might be wrong. In addition, there are some specific questions that need to be clarified.

(1) During the reaction, how long can the specific shape of Pd NPs, i.e., cubes and octahedrons, keep? Could you give more low magnification images after the reaction? It's better to provide the statistical information of the dynamical changes during the reaction of the Pd NPs .

(2) When comparing the activity of Pd cubes and Pd octahedrons in Figure 1, did the authors consider the amount of catalysts used during in situ experiments and also the distribution of the catalysts since the temperature of the heat chip is not homogeneous?

(3) To remove the residual surfactants, the catalysts were heated to 300 °C under a flow of Ar for 30 min and followed by heat treatment in 20% O₂ and 80% He for 10 min. How to prove the Pd NPs are clean

enough. Because the synthesis methods of Pd cubes and Pd octahedrons are different. If the residual surfactants are not removed completely, they may influence the activity of Pd NPs.

(4) For Figure 1D, can the authors zoom in the oscillation of gas partial pressures and measured temperature? It's not clear to see the relationship between them at present.

(5) For Figure 2, it's better to mark the time of the images (C and D) on the curves (A and B).

(6) How to know {120} and {311} facets from Figure 2D?

(7) For Figure 3, the authors just gave the specific 102 s, 204 s, 326 s, how about other similar moments near these times? More images should be showed in SI. And again, it's better to mark the corresponding images on the curves.

Response to Reviewer #1:

Ghosh et al. presented an interesting study to explore the underlying mechanism of oscillatory behavior of CO oxidation on Pd nanoparticles. The in-situ experimental and the DFT-based multiscale simulations showed this phenomenon is induced by the reconstruction of Pd nanoparticles owing to the change of the adsorbates' coverage. The paper is well-written and the demonstration is very clear. I think this paper meets the criteria of Nature Communications once the authors clarify the following issues.

We appreciate the reviewer's comments about the quality of our study and address his/her comments below.

(1) The experiments indicated the nanocubes did not exhibit the oscillation behavior as the nano-octahedrons. And the simulation gives the equilibrium structure at the specific temperature. Once the nanocubes and the nano-octahedrons were exposed in the same conditions for a long time, such as 400 °C, CO:O₂ = 0.5, would they transform to similar geometries and exhibit similar oscillatory behaviors?

Yes, the nanocubes will transform into truncated nanocubes under prolonged exposure to a reactive environment. For example, Supplementary Figure 16 shows how nanocubes change into truncated nanocubes after spending 6 h at $p_{\text{CO}}/p_{\text{O}_2} \approx 0.5$ and $T = 400$ °C, CO:O₂ = 0.5. As a consequence of this transformation, these newly formed truncated nanocubes can undergo reaction oscillations, as shown in Supplementary Figure 17.

Following the reviewer's comment, we have now explicitly discussed this point in our manuscript: *"Note that even during prolonged CO oxidation reaction, we observe the truncation of nanocubes because of the O₂ in the gas mixture. For example, under the O₂-rich reaction conditions ($p_{\text{CO}}/p_{\text{O}_2} \approx 0.5$), nanocubes transformed into truncated nanocubes with after approx. 6 h (Supplementary Figure 16) and started to display oscillatory reactivity which is not observed in nanocubes (Supplementary Figure 17)."*

Regarding nano-octahedrons, they are close to equilibrium structure (Figure 5b), hence they do not undergo considerable structural changes under similar conditions and retain their shape even after a prolonged reaction (Supplementary Figure 7).

(2) Why did the nanocubes not show the oscillation? Is the activity of nanocubes too high to show the oscillation? Are the low activity surfaces necessary for the oscillation?

The nanocubes do not show oscillations because both high- and low-activity surfaces (*i.e.*, high- and low-index facets) are necessary for oscillations to occur, and the nanocubes are dominantly {100}-faceted with very few higher-index facets at the edges and corners. Furthermore, these kinetically trapped cubic shape does not easily restructure, unlike the nano-octahedron and truncated nanocubes, which are much closer to their {111}-faceted equilibrium shapes as shown in Figure 5b. In other words, dynamic oscillations take place only when there are two stable states and the system is able to switch between them easily.

No, the activity of nanocubes is not too high. Even the active nanocubes have much lower activity than the other two nanoparticle types (see Figure 1c vs. Figures 1d and 4b).

Yes, both high and low-activity surfaces are necessary because the main cause of the oscillations is restructuring between low- and high-index-faceted surfaces.

(3) The oscillated CO oxidation is a collective phenomenon. Clearly, different nanoparticles may stay at the different stages of the structure reconstruction. In this work, the authors give the observation of a single nanoparticle. I wonder if this oscillatory behavior is persistent during long-time reactions or only appears at the beginning stage. After long time operations, will the nanoparticles synchronize their structure reconstructions to show the stronger oscillation, or offset the individual effects to show the weaker oscillation?

Yes, during the oscillatory reactions, nanoparticles restructure in tandem with their reactivity, oscillating between low- and high-activity states in a synchronous manner. Supplementary Figures 18–19 show how several truncated nanocubes restructure synchronously during two different oscillatory reactions. Supplementary Figure 12 shows that these oscillations get stronger with time until the oscillation amplitude reaches a peak within few cycles.

(4) Can the authors provide the near-surface lattice distance in Figure 3 to show the oxidation contents of Pd nanoparticle during the CO oxidation?

The near-surface lattice distance in Figure 3 is 0.197 nm, which corresponds to the lattice spacing of Pd {100} planes (0.194 nm) as labeled by yellow arrows in Figure 3e.

(5) Did the total pressure keep at 1 bar for all reactions?

Yes, the total pressure was 760 Torr or 1 bar for all our operando TEM reactions, which is now explicitly mentioned in “*Operando TEM experiments*” subsection of the methods section.

Response to Reviewer #2:

The paper addresses a very important topic in catalysis, that is, the nature of the active site in reaction conditions. In particular, the authors report pieces of evidence for the periodic structural changes in Pd NP during CO oxidation. They link the CO oxidation oscillatory behavior with periodic morphological changes of the NP. This is proved experimentally mostly by correlating operando-TEM with the evolution of CO/CO₂/O₂ during time. On one side, the paper is well written and reports very interesting results. Results are novel as they provide an alternative explanation of the behavior already reported in the literature. On the other side, I have some points to be clarified that should be taken into consideration before publication:

We thank the reviewer for his/her comments, which we address below.

(1) the difference between the behavior of nanocube NP and nano-octahedron NP is related to the presence of (111) facets. As far as I understand from the paper, this is first considered as the explanation because it is known (page 3 lines 107-109) that the reaction has lower activation energy on (111) than on (100). However, later (page 6 lines 149-151) the oscillatory behavior of the octahedron NP is explained via the formation of high index facets. I found this quite confusing and not coherent. A clarification is required.

As we stated on page 3, the light-off temperature is lower for nano-octahedrons because of its lower activation energy. However, once the light-off occurs, a nano-octahedron restructures, creating high-index facets with under-coordinated sites at its edges and corners, increasing the number of active sites responsible for further increase in CO conversion. Note that light-off does not require restructuring, and CO conversion can start on {111} facet. However, further restructuring into high-index faceted

from {111}-terminated nano-octahedron (*i.e.*, rounding) causes catalyst switch from low-activity (still active) to high-activity state. Hence, while only {111} facets can account for light-off (*i.e.*, going from fully inactive to active state), switching between low- and high-activity states requires restructuring of the nanoparticle.

(2) An important effect is attributed to CO coverage. Are there available any IR experiments to prove the relative presence of these adsorbates at the different T and CO/O₂ ratios?

Yes, there are ample reports using IR experiments that show CO coverage increases as the amount of CO increases (*i.e.*, for higher CO/O₂ ratio) in the reactive environment (Ref. 19 and Ref. 20). Our results showing an upward shift in ignition temperature with an increase in CO/O₂ ratio is in line with these earlier studies.

Following the reviewer's comment, we have now included Ref. 19 and Ref. 20.

(3) CO could bind differently on the different terminations thus inhibiting the reaction on the different facets at different T .

The reviewer correctly points out that CO binds differently on different facets. That is why nanocubes (with {100} termination) are inactive at low temperatures while nano-octahedrons (with {111} termination) are highly active.

This is also captured in our DFT and thermodynamic calculations showing that both the CO-coverage and the change in the coverage with temperature are different on the different facets, as shown in Supplementary Figures 30–31. Both of these account for a distinct behavior of the calculated surface energies of different facets at varying temperatures (Supplementary Figure 27), and ultimately dictate the overall shape of the nanoparticle and its activity (Figure 5 and Supplementary Figure 28).

(4) A point not clear to me is the origin of the oscillation. It can only start from the NP, since the feed is being kept constant. Comment on it should be important to better clarify the phenomenon.

The link between the structural changes and oscillations in activity under constant reaction conditions can be understood by recognizing that the metal surface rarely consists of both CO and O adsorbates, which are required for the reaction to proceed. (Ref. 19) The Pd surface is, in general, dominated by adsorbed CO molecules regardless of the gas composition, as this molecule binds more strongly than O (Refs. 52–53). Once light-off occurs, the energy released by CO oxidation will sustain the reaction, leading to ignition (Ref. 2). However, this is true only if CO molecules can combine with O to produce CO₂. As mentioned in response to a comment (3) of the same reviewer, the binding strength of CO is different on different facets. We can reasonably assume that any reaction will initiate at the corners of a catalyst particle since they are expected to be the high activity sites (Refs. 52–53), but the further reaction cannot occur if the lower index facets remain poisoned by CO. In this case, the subsequent conversion will be quenched, and the exposed catalyst surface becomes re-populated with CO molecules, thereby setting up the oscillatory phenomena. This explanation is also consistent with the increase in oscillation frequency with an increase in temperature (Supplementary Figure 17), where the desorption of CO becomes favorable, creating sites for reaction with O (Supplementary Figures 30–31).

Following the reviewer's comments, we elaborated this point in more detail on pages 10–11 of the revised manuscript.

(5) It is of extreme importance to provide evidence that the experimental rig is not affected by mass/heat transfer limitations. These aspects should be reported in the paper (with details in the SI).

Regarding mass transport:

a) If the oscillations were due to limited mass transport then, we would expect significant changes in the amount of the reactants during the oscillations; however, for all our experiments, there is always a sufficient amount of CO and O₂ in the gas cell, as evidenced by multiple experiments under different reaction conditions in similar cells (Supplementary Figures 17 and 20).

b) Oscillations are not seen in the active Pd nanocubes (albeit at lower conversion rates) under similar conditions.

c) Oscillations are only specific to Pd nano-octahedrons and truncated nanocubes. Our earlier studies with randomly shaped Pt and Rh nanoparticles (Ref. 2) and Pt–Ni alloy nanoparticles (Tan *et al.*, *Adv. Func. Mater.* 29, 1903242 (2019) under similar conditions, did not display oscillatory reactions.

Hence, we conclude that limited mass transport is not the cause of the observed oscillations in our Pd nanoparticles.

Regarding heat transfer:

a) Drop-casted and annealed nanoparticles are in full contact with SiN_x film, which has sufficiently high thermal conductivity ($\kappa_{\text{SiN}_x} \approx 4.9 \text{ W m}^{-1} \text{ K}^{-1}$), and heat can be conducted away very efficiently, in timescales much less than those of the oscillation frequencies.

b) If the oscillations were due to limited thermal transport, then the oscillation frequency would not strongly depend on CO amount; however, oscillations do depend on the amount of CO, as shown in Supplementary Figure 20.

c) As mentioned earlier, our earlier studies in a similar gas cell but on different nanoparticle systems do not display oscillatory reaction suggesting that the observed process is not due to limited thermal transport.

(6) the theoretical analysis is of interest and important to support the interpretation of the results. However, there are recent papers that show that Wulff construction is not always a good model (e.g., the system needs time to reach such equilibrium - *Cat. Sci. Tech.* 8 (2018) 3493 and *ACS Catal.* 10 (2020) 6149). Are the periods of the oscillations long enough to allow the NPs to reach their equilibrium condition? Comment on this would be of added value.

The reviewer raises a very important point about the limitations of modeling. Here, we emphasize that the goal of our computational modeling is to rationalize the trends observed experimentally. We do so by inspecting the energetics of stable equilibrium states. True, this by itself does not allow a prediction of the rate of change. However, our experimental measurements clearly show a reshaping of the nanoparticles, a clear indication that the rate of change is within the timescale of the periodicity. Therefore, combining the modeling and experiments show that the system reshapes towards the identified stable equilibrium states.

We added the following sentence on pages 9–10 to emphasize this point: “*The transient nature of the NPs during the reactions indicates a swift response to external driving forces, which we aim to rationalize focusing on endpoints of this process, i.e., equilibrium states, providing accurate nanoparticle models.*”

Response to Reviewer #3:

This work studied the periodic round-to-flat transitions of Pd NPs during CO oxidation reaction by using operando transmission electron microscopy. They used shape-controlled NPs and monitored structural changes, reaction temperature and heater power, and output gas compositions at constant reaction conditions. It is an interesting work to see periodic structural changes in Pd nanoparticles during oscillatory CO oxidation, although it is similar to what was observed for Pt nanoparticles during CO oxidation [Ref. 3 in the manuscript]. Nevertheless, the authors have conducted a systematic study in order to obtain insights into Pd catalyzed CO oxidation. However, to explain what happened in this reaction, the authors seem to ignore the fact that Pd was partially oxidized at such a high temperature [refer to Fig. 4C and attached videos, the materials with weaker contrast around Pd NPs should be PdO_x], which makes the conclusion questionable. In my experience, at such a high temperature with O₂, the Pd should be oxidized. If the Pd NPs were oxidized, the conclusion that “periodic restructuring of NP catalysts is the root cause of oscillation in reactivity for both O₂- and CO-rich cases and not caused by the oxidation of the surface as previously proposed” and the modeling might be wrong. In addition, there are some specific questions that need to be clarified.

We thank the reviewer for her/his kind remarks.

We did consider the possibility of Pd oxidation having an effect on the reaction but do not believe that this would be the case:

See page 9 of the revised manuscript that elaborates our point: “*Noteworthy, the materials with weaker contrast around the NP in Figure 4c may be an indication of PdO_x; however, their structures do not change during the activity changes coinciding with the –flat–round–flat– structural oscillation of the NP. Moreover, many earlier reports suggest that Pd can be partially oxidized during the reaction, but the oxide phase does not contribute significantly to the observed reactivity.^{19,40} Also, note that, the occurrence of the weaker contrast material is infrequent while the –flat–round–flat– structural change is ever-present. Therefore, it is improbable that the weaker contrast materials have a role in causing the oscillation.*”

First, only structural oscillations occur concurrently with the oscillating reactions.

Second, most of the nanoparticles do not have any (visually) detectable oxide on their surfaces (*i.e.*, weaker contrast material patches on the nanoparticles as the one shown in Figure 4c), but they still do exhibit oscillatory reactivity.

Third, while oxides may be present on few nanoparticles, their amount does not change concurrently with the activity during the oscillations; only the structure of the nanoparticles changes in tandem with the activity (see Supplementary Figures 10–15), suggesting that oscillations are unlikely to be due to oxides (Ref. 20, Ref. 40).

(1) During the reaction, how long can the specific shape of Pd NPs, *i.e.*, cubes and octahedrons, keep? Could you give more low magnification images after the reaction? It’s better to provide the statistical information of the dynamical changes during the reaction of the Pd NPs.

Under typical reaction conditions, nanocubes, nano-octahedrons, and truncated retain their original shape after typical experimental timescales of <6 h (Supplementary Figures 6 and 16), >8 h (Supplementary Figure 7), >3 h (Supplementary Figure 15), respectively.

We now explicitly refer to these new figures in the main text: “*Note that both cubic and octahedral NPs retain their original shape even after the conversion reactions that last several hours, indicating that all structural changes occurring during the reaction are transient (see Supplementary Figures 6–7 and 16).*”

Following the reviewer's comment, more images showing the dynamic changes of NPs have now been added in Supplementary Figures 13–15.

(2) When comparing the activity of Pd cubes and Pd octahedrons in Figure 1, did the authors consider the amount of catalysts used during in situ experiments and also the distribution of the catalysts since the temperature of the heat chip is not homogeneous?

To avoid any discrepancy in the measurements due to the number of the nanoparticles in the active area (*i.e.*, heater area), for each experiment, we drop-casted 1 μL of nanoparticles with similar concentrations. Next, we also verified that the coverage in active area is more or less similar for each experiment by low-magnification screening prior to the experiments. We now explicitly mention this in “*Operando TEM experiments*” subsection of the methods section.

(3) To remove the residual surfactants, the catalysts were heated to 300 °C under a flow of Ar for 30 min and followed by heat treatment in 20% O₂ and 80% He for 10 min. How to prove the Pd NPs are clean enough? Because the synthesis methods of Pd cubes and Pd octahedrons are different. If the residual surfactants are not removed completely, they may influence the activity of Pd NPs.

We have used a ubiquitous and effective method to remove surfactants; washing nanoparticles by centrifugation followed by heating them at high temperatures (*e.g.*, 300 °C) (Ref. 45).

At 300 °C, residual CTAB on the nanocubes completely decomposes (Ref. 46). That is why to fully remove CTAB after centrifugation, we prebake the nanoparticles at 300 °C before any catalytic reaction (Ref. 48).

PVP surfactant (used in the synthesis of nano-octahedron) can effectively be removed by heat treatment in 20% O₂ and 80% He, at 200–300 °C *via* combustion and/or decomposition of PVP (Ref. 47).

Furthermore, all our experiments are conducted at temperatures much higher than 300 °C, and the presence of any residual surfactant on nanoparticles is highly unlikely.

The references mentioned here are now included as refs. 45-48 in the revised manuscript.

(4) For Figure 1d, can the authors zoom in on the oscillation of partial gas pressures and measured temperature? It's not clear to see the relationship between them at present.

A zoomed-in version of the graph has now been presented separately as Supplementary Figure 4.

(5) For Figure 2, it's better to mark the time of the images (c and d) on the curves (a and b).

Following the reviewer's comment, we now labeled the timepoints corresponding to the image series in Figure 2c–d by black arrows in Figure 2b.

(6) How to know {120} and {311} facets from Figure 2d?

To help in the visualization of the high-index facets, *i.e.*, {120} and {311}, we added a figure of a fictitious nanoparticle (with all of its planes labeled) as Supplementary Figure 33, which depicts a truncated nanocube used in this work.

(7) For Figure 3, the authors just gave the specific 102 s, 204 s, 326 s; how about other similar moments near these times? More images should be shown in SI. And again, it's better to mark the corresponding images on the curves.

Following the reviewer's suggestions, more images have now been added as Supplementary Figure 11 in the revised Supplementary Information.

REVIEWER COMMENTS

Reviewer #1 (Remarks to the Author):

The authors have well addressed my concerns and elucidated a self-consistent mechanism of the periodic structure change of Pd NP under CO oxidation reaction. I recommend the publication of this manuscript in Nature Communications as is.

Reviewer #2 (Remarks to the Author):

The points raised in my previous assessment have been addressed in detail in the revised version of the manuscript. The paper can be published in its present form.

Reviewer #3 (Remarks to the Author):

Thanks for the response of the authors. It answered most of my questions. However, I am still unclear about the PdOx. During the oscillation of CO oxidation, why some of the Pd NPs were oxidized while some were not? The most important conclusion of this work is that the restructuring of the surface of the Pd NPs is the origin of the CO oxidation oscillation. In my experience, the changes of the catalyst, like shape, composition, always have some effects on the catalytic reaction. Here, the big changes of the catalyst composition that the Pd was oxidized, it must have some effects on the whole reaction. It's better to study and clarify the role of the PdOx during the whole oscillation of CO oxidation. By the way, in Figure 4c, the transformation of the NP corner between the round and the flat is not obvious.

About the origin of the reshaping in Page10-11, it's also not clear to me. How does the oscillation proceed? The reaction will initiate at the corners of a catalyst particle since they are expected to be the high active sites. As the reaction happened, the heat released from the reaction would further speed the reaction. In other words, the temperature increased with the heat released. The corners should remain as higher index facets. Then the reaction can not oscillate because of continuous high activity. But the authors said that "However, the reaction cannot proceed further if the lower index facets remain poisoned by CO". If the lower index facets can not catalyze the reaction, the higher index facets should proceed the reaction continuously. The authors should clarify the whole process more clearly. Furthermore, there is another possibility that the shape of the nanoparticle changes because of partial pressure of the CO and O₂. For example, when the partial pressure of CO increases, the Pd NPs prefer to expose (100) facet. While when the partial pressure of O₂ is higher, the Pd NPs prefer to expose higher index facets that the NPs look more round. In this way, the reshaping of the Pd NPs is the result of the changes of the gas environment, not the origin of the oscillation of the CO oxidation. The authors should clarify more clearly and directly about this.

Response to Reviewer #1:

The authors have well addressed my concerns and elucidated a self-consistent mechanism of the periodic structure change of Pd NP under CO oxidation reaction. I recommend the publication of this manuscript in Nature Communications as is.

We thank the reviewer for recommending the publication of our manuscript in Nature Communications.

Response to Reviewer #2:

The points raised in my previous assessment have been addressed in detail in the revised version of the manuscript. The paper can be published in its present form.

We thank the reviewer for recommending the publication of our manuscript in Nature Communications.

Response to Reviewer #3:

Thanks for the response of the authors. It answered most of my questions. However, I am still unclear about the PdOx. During the oscillation of CO oxidation, why some of the Pd NPs were oxidized while some were not? The most important conclusion of this work is that the restructuring of the surface of the Pd NPs is the origin of the CO oxidation oscillation. In my experience, the changes of the catalyst, like shape, composition, always have some effects on the catalytic reaction. Here, the big changes of the catalyst composition that the Pd was oxidized, it must have some effects on the whole reaction. It's better to study and clarify the role of the PdOx during the whole oscillation of CO oxidation. By the way, in Figure 4c, the transformation of the NP corner between the round and the flat is not obvious.

Following the reviewer's comment, we performed a control experiment in which we intentionally oxidized the NPs under an oxygen-only environment and used these NPs for CO oxidation study to show that occasionally the surface of the NPs may have weak-contrast carbonaceous residues as that in Figure 4c but not PdO. Our new results are summarized in new Supplementary Figures 21–22 and in the following changes to the manuscript on page 9: *“A careful examination of the NP surface in Figure 4c reveals the presence of a low-contrast amorphous material, which we attribute to carbonaceous residues left from the NPs synthesis because of the following two reasons. First, these residues were only observed on rare occasions (for example, none of the NPs shown in Supplementary Figures 18–19 display any surface residues). Second, the structure of residues did not change during the entire reaction. Nevertheless, to ensure that these surface materials are not PdO, which might have formed during the reaction,^{19,20} we first carefully oxidized the surface of the Pd nanocubes (in 20% O₂ and 80% He atmosphere at 460 °C) and found that the image contrast from the crystalline surface PdO (Supplementary Figure 21b) is much stronger than from the surface residues in Figure 4c. Moreover, this oxide reduces and vanishes under the typical reaction condition ($p_{CO}/p_{O_2} \approx 0.8$ at 460 °C) (Supplementary Figure 21c). In any case, because PdO rapidly reduces into metallic Pd in the presence of CO (Supplementary Figure 21c), any initial surface oxide that may be present on the surface of our Pd NPs does not affect their reactivity (i.e., the NPs retain their oscillatory reactivity) (Supplementary Figure 22).”*

About the origin of the reshaping in Page10-11, it's also not clear to me. How does the oscillation proceed? The reaction will initiate at the corners of a catalyst particle since they are expected to be the

high active sites. As the reaction happened, the heat released from the reaction would further speed the reaction. In other words, the temperature increased with the heat released. The corners should remain as higher index facets. Then the reaction can not oscillate because of continuous high activity. But the authors said that “However, the reaction cannot proceed further if the lower index facets remain poisoned by CO”. If the lower index facets can not catalyze the reaction, the higher index facets should proceed the reaction continuously. The authors should clarify the whole process more clearly. Furthermore, there is another possibility that the shape of the nanoparticle changes because of partial pressure of the CO and O₂. For example, when the partial pressure of CO increases, the Pd NPs prefer to expose (100) facet. While when the partial pressure of O₂ is higher, the Pd NPs prefer to expose higher index facets that the NPs look more round. In this way, the reshaping of the Pd NPs is the result of the changes of the gas environment, not the origin of the oscillation of the CO oxidation. The authors should clarify more clearly and directly about this.

It is our current hypothesis that the gas environment and structural changes of NPs are coupled instead of one driving another (*i.e.*, change in activity affects the local gaseous environment of the catalysts, which in turn affects catalysts' state). Page 7 of the main text discusses that one of the prevailing hypotheses for how such synchronized oscillations in reactivity can arise between separated catalyst particles is that the oscillations are coupled through local changes in the gas composition due to the reactivity fluctuations. Hence, what we described is specifically how reactivity fluctuations can be initiated in the "first" catalyst particle, but the propagation of the fluctuations onwards to its neighboring particles is likely through a mechanism described by the reviewer. In this case, the changes in local gas composition drive structural changes in more particles, which in turn leads to further changes in the gas composition that begin to quench the reactivity, setting up the observed oscillatory behavior. We emphasize here that the two processes cannot be decoupled and rationalized independently.

REVIEWERS' COMMENTS

Reviewer #3 (Remarks to the Author):

The authors have conducted additional experiments to address the comments raised by the reviewer, who wants to see some different things (explanation). At the end, it seems that they just followed the scenario which was proposed in Nat. Mater. 13, 884-890 (2014). The only difference is that Pt is used in that paper and Pd is in the current work. The authors did not make it clear in the introduction part and even did not cite it when talking about the oscillation during reaction. Moreover, they stated in the conclusion part that "we anticipate that xxx can be extended to many other systems", "...More generally, xxx". This is over claimed as compared with what reported in that Nat. Mater. paper.

The authors should rewrite these parts and specify the contribution of the Nat Mater paper, in order to give a clear picture for the readers to appreciate these two works. In addition, "This restructuring causes spontaneous oscillations in the conversion xxx" and "xxx is the root cause of oscillation" are not appropriate. The authors did not address the issue as the reviewer raised last round and just mentioned "the two processes cannot be decoupled". At present, they did not have solid evidence to claim which one is the cause and which one is the result. In a word, overclaimed description should be seriously revised before publication.

Response to Reviewer #3:

The authors have conducted additional experiments to address the comments raised by the reviewer, who wants to see some different things (explanation). At the end, it seems that they just followed the scenario which was proposed in Nat. Mater. 13, 884-890(2014). The only difference is that Pt is used in that paper and Pd is in the current work. The authors did not make it clear in the introduction part and even did not cite it when talking about the oscillation during reaction. Moreover, they stated in the conclusion part that "we anticipate that xxx can be extended to many other systems", "...More generally, xxx". This is over claimed as compared with what reported in that Nat. Mater. paper.

The authors should rewrite these parts and specify the contribution of the Nat Mater paper, in order to give a clear picture for the readers to appreciate these two works. In addition, "This restructuring causes spontaneous oscillations in the conversion xxx" and "xxx is the root cause of oscillation" are not appropriate. The authors did not address the issue as the reviewer raised last round and just mentioned "the two processes cannot be decoupled". At present, they did not have solid evidence to claim which one is the cause and which one is the result. In a word, over claimed description should be seriously revised before publication.

Regarding the use of a different material only:

We kindly disagree with the reviewer that the current work just follows the scenario of Nat. Mater. 13, 884-890 (2014) for the following reasons:

a) Nat. Mater. 13, 884-890 (2014) uses randomly shaped sphere-like NPs; therefore, authors could not identify the role of different facets and edges between the facets both on activity and the oscillations. In our study, we accomplish exactly this by systematically varying the shape of NPs and examining them under different reaction conditions.

b) Nat. Mater. 13, 884-890 (2014) claims that "*a more faceted shape is expected*" for low CO content (*i.e.*, resulting in more or less equal O binding both to steps and facets). Our study, where we systematically varied $p_{\text{CO}}/p_{\text{O}_2}$, contrasts this point and shows that facet stabilization is due to high CO content (consistent with our earlier study, Nat. Commun. 11, 2133 (2020)).

c) Unlike Nat. Mater. 13, 884-890 (2014), we present clear evidence that oscillations across different particles in the reaction chamber occur synchronously, which is critical for sustainable oscillations. Moreover, Nat. Mater. 13, 884-890 (2014) notes that the oscillations occur at the exit of the reaction cells (*i.e.*, observations are strongly influenced by the geometry of the experimental setup). In our case, observations are not affected by the setup geometry, and we see the oscillation in all observable areas of the reactor.

Regarding the citation of Nat. Mater. 13, 884-890 (2014):

This paper has been cited three times throughout the manuscript. Nevertheless, we now explicitly mention it yet again in the 3rd paragraph of the introduction:

"Moreover, a recent operando TEM study of oscillatory reaction in Pt NPs by Vendelbo et al. directly shows that structural changes and reactivity are correlated."³

Regarding the statement in the conclusion section:

In "*We anticipate that this dynamic structure-dependent activation–deactivation behavior in Pd NPs at constant reaction conditions extends to many other catalyst systems.*", the verb *anticipate* reflects our speculation of what lies ahead, and because the choice of the word

“anticipate” makes it clear that this is speculation/prediction/expectation, it cannot be an overclaim by definition. Hence, we would like to keep the wording unchanged.

In “*More generally, our approach to studying the interaction between catalyst surface and reactants with the sub-nanometre resolution is critical for identifying and eliminating different deactivation modes and will be important in guiding the design of future high-performance catalysts.*”, simply emphasizes the role of operando studies in gaining insight into the nanoscale details of nanocatalysts and by no means meant to be an overclaim. For example, in recent papers, other authors reiterate a similar point:

Ek *et al.*, *Nat. Commun.* **8**, 305 (2017): “*Thus, the ability to observe oxide catalysts at atomic resolution under meaningful chemical conditions is essential for developing atomic-scale understanding of oxide-catalyzed chemical reactions, and of the redox chemistry of nanostructured oxide materials in general.*”

Helveg *et al.*, *Micron* **68**, 176-185 (2015): “*To describe the catalytic functionality, it is therefore essential to have information about the size, shape, and surface structure of the catalyst at the atomic level. With recent advancements in aberration-corrected electron optics, as well as data acquisition and analysis schemes, (scanning) transmission electron microscopy techniques now enable observations of catalysts and other nanomaterials with a substantially reduced contrast delocalization, an enhanced resolution as well as a sensitivity that reaches the ultimate single-atom level.*”

Crozier and Hansen, *MRS Bulletin* **40**, 38-45 (2015): “*This approach provides a powerful technique for exploring how different in situ treatments such as reduction and oxidation can change the activity of a catalyst. Operando TEM will allow atomic-level structure to be correlated with changes in relative reaction kinetics and will greatly enhance the determination of structure–reactivity relations.*”

Hence, we would like to keep the wording in our sentence unchanged.

Regarding the cause of oscillation:

Oscillations in structure and activity are coupled, and one feeds on the other, meaning that once triggered, the change in structure causes the change in activity, and this change in activity results in a change in the structure, and the cycle repeats. Hence, from a semantics point of view, the following sentence is still correct: “*This restructuring causes spontaneous oscillations in the conversion of CO to CO₂ under constant reaction conditions.*”

When we say that “*We demonstrated that periodic restructuring of Pd NP catalysts is the root cause of oscillation in reactivity for both O₂- and CO-rich cases, and these oscillations are not caused by the oxidation of the surface as previously proposed,*” we mean to highlight the fact that we have not detected any surface oxidation during the oscillations. Nevertheless, following the reviewer’s suggestion, we toned down the statement in the revised manuscript by replacing “*are not caused*” with “*are unlikely to be caused.*”